# PKD: General Distillation Framework for Object Detectors via Pearson Correlation Coefficient

**Weihan Cao**[1,3]     **Yifan Zhang**[1] *     **Jianfei Gao**[2]     **Anda Cheng**[1,3]
**Ke Cheng**[1,3]     **Jian Cheng**[1]

NLPR & AIRIA, Institute of Automation, Chinese Academy of Sciences[1]
Shanghai AI Laboratory[2]
School of Artificial Intelligence, University of Chinese Academy of Sciences[3]
caoweihan2020@ia.ac.cn
{yfzhang, jcheng}@nlpr.ia.ac.cn
gaojianfei@pjlab.org.cn
{chenganda, chengke}2017@ia.ac.cn

## Abstract

Knowledge distillation(KD) is a widely-used technique to train compact models in object detection. However, there is still a lack of study on how to distill between heterogeneous detectors. In this paper, we empirically find that better FPN features from a heterogeneous teacher detector can help the student although their detection heads and label assignments are different. However, directly aligning the feature maps to distill detectors suffers from two problems. First, the difference in feature magnitude between the teacher and the student could enforce overly strict constraints on the student. Second, the FPN stages and channels with large feature magnitude from the teacher model could dominate the gradient of distillation loss, which will overwhelm the effects of other features in KD and introduce much noise. To address the above issues, we propose to imitate features with Pearson Correlation Coefficient to focus on the relational information from the teacher and relax constraints on the magnitude of the features. Our method consistently outperforms the existing detection KD methods and works for both homogeneous and heterogeneous student-teacher pairs. Furthermore, it converges faster. With a powerful MaskRCNN-Swin detector as the teacher, ResNet-50 based RetinaNet and FCOS achieve 41.5% and 43.9% $mAP$ on COCO2017, which are 4.1% and 4.8% higher than the baseline, respectively. Our implementation is available at https://github.com/open-mmlab/mmrazor.

## 1 Introduction

Knowledge distillation(KD) is a widely-used technique to train compact models in object detection. However, there is still a lack of study on how to distill between heterogeneous detectors. Most previous works [40, 35, 10, 20, 13] rely on detector-specific designs and can only be applied to homogeneous detectors. [44, 50] conduct experiments on detectors with heterogeneous backbones, but detectors with heterogeneous detection heads and different label assignments are always omitted. Object detection is developing rapidly and algorithms with better performance are proposed continuously. Nevertheless, it is not easy to change detectors frequently in terms of stability in practical applications. Furthermore, in some scenarios, only detectors with a specific architecture can be deployed due to hardware limitations (e.g., two-stage detectors are hard to deploy), while most powerful teachers belong to different categories. Thus, it is promising if knowledge distillation can be conducted

---

*Corresponding author

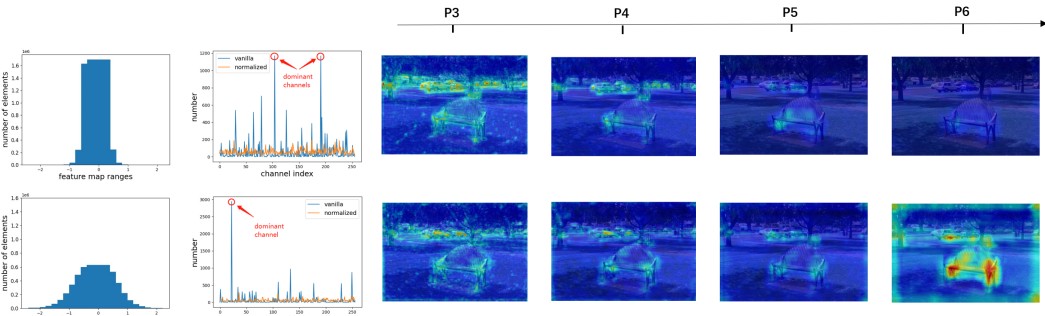

Figure 1: An example to illustrate the problems suffered by directly aligning the feature maps. The first and second lines correspond to the teacher and the student. Left: Feature magnitude of the teacher and the student. Middle: Dominant channels in FPN stage 'P3'. Let $s_{l,u,v} \in \mathbb{R}^C$ denote the feature vector located in pixel (u, v) from $l$-th FPN stage and omit $l$ for clarity. Then $number_i = \sum_{u,v} \mathbb{1}[\arg\max_c s_{u,v}^{(c)} = i]$ where $i$ is the channel index. We define channels with a larger $number$ as dominant channels. Right: Visualization of the activation patterns from the teacher and the student. More visualization is provided in the Appendix.

between heterogeneous detector pairs. In addition, current distillation methods, such as [44, 47], usually introduce several complementary loss functions to further improve their performance, so several hyper-parameters are used to adjust the contribution of each loss function which heavily affects their abilities to transfer to other datasets.

In this paper, we first empirically verify that FPN feature imitation can distill knowledge successfully even though the student-teacher detector pairs are heterogeneous. However, directly minimizing the Mean Square Error (MSE) between features of the teacher and the student leads to sub-optimal results, with results shown in Table 5. Similar conclusions are drawn in [6, 13, 50, 44]. In order to explore the limitations of MSE, we elaborately visualize the FPN feature responses of the teacher and student detectors, as shown in Figure 1. Specifically, for an output feature $s_l \in \mathbb{R}^{C \times H \times W}$ from $l$-th FPN stage, we select the maximum value in the dimension $C$ at each pixel and obtain a 2-D matrix. Then we normalize the values to 0-255 according to the maximum and minimum values of $l$ 2-D matrices. Through these comparisons, we obtain the following observations:

**(1) The feature value magnitudes of the teacher and the student are different, especially for heterogeneous detectors, as shown in Figure 1 (left).** So directly aligning the feature maps between the teacher and the student may enforce overly strict constraints and do harm to the student.

**(2) The values of several FPN stages are larger than the others, as shown in Figure 1 (right).** It is obvious that FPN stage 'P6' of the teacher is less activated than FPN stage 'P3'. However, for detectors such as RetinaNet [24] and FCOS [37], all FPN stages share the same detection head. Hence, FPN stages with larger values could dominate the gradient of the distillation loss, which will overwhelm the effects of other features in KD and lead to sub-optimal results.

**(3) The values of some channels are significantly larger than the others, as shown in Figure 1 (middle).** However, it is mentioned in [35, 48] that the less activated features are still practical for distillation. If they are not correctly balanced, the small gradients produced by these less activated channels can be drowned in the large gradients produced by the dominant ones, thus limiting further refinement. Furthermore, from the first column of Figure 1 (right), we observe that there is much noise in the object-irrelevant area, because the values of the pixels in certain channels are significantly larger than those in other channels, thus being visualized in the figure. However, these pixels may be noise. Hence, directly imitating the feature maps may introduce much noise.

According to these observations, we propose **K**nowledge **D**istillation via **P**earson Correlation Coefficient (PKD) shown in Figure 2, which focuses on the linear correlation between features of the teacher and the student. To remove the negative influences of magnitude difference between the teacher-student detector pair and within the detector among different FPN stages and channels, we first normalize the feature maps to have zero mean and unit variances and minimize the MSE loss between the normalized features. Mathematically, it is equivalent to firstly calculating the Pearson

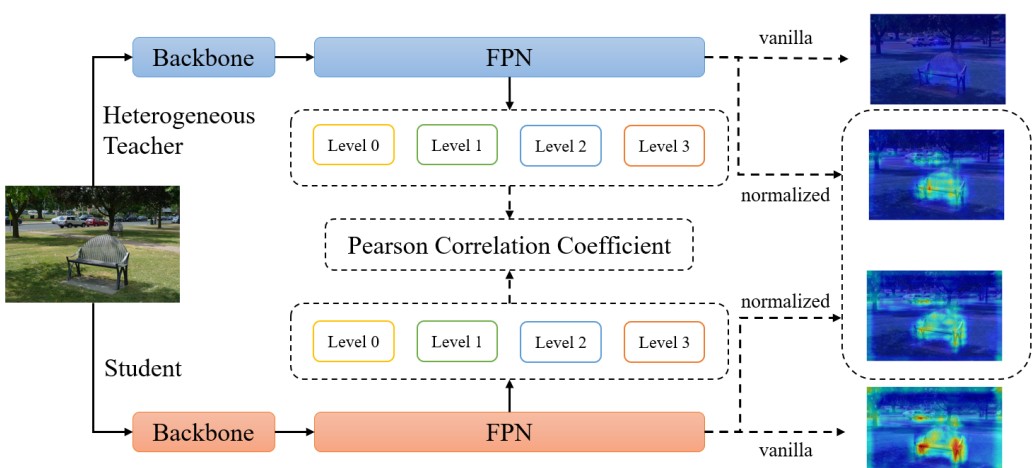

Figure 2: Overview of the proposed distillation via Pearson Correlation Coefficient (PKD). To demonstrate the effectiveness of PKD, we visualize the discord activation patterns of pre-normalized and post-normalized FPN features before distillation. The normalization mechanism bridges the gap between the activation patterns of the student and the teacher.

Correlation Coefficient ($r$) [2] between two original feature vectors, and then using $1 - r$ as the feature imitation loss.

Compared with the previous methods, our method offers the following advantages. First, as the distillation loss is calculated just on FPN features, it can be easily applied to heterogeneous detector pairs, including models with heterogeneous backbones, heterogeneous detection heads and different training strategies such as label assignment. Second, as there is no need to forward the teacher's detection head, the training time can be reduced significantly, especially for those models with cascaded heads. Besides, PKD converges faster than previous methods. Last but not least, it has only one hyper-parameter - distillation loss weight and is not sensitive to it. So it can be easily applied to other datasets. We conduct extensive experiments to verify the significant performance boosts brought by our approach on COCO dataset [25]. Using the same two-stage detector as the teacher, ResNet50 based RetinaNet [24] and FCOS [24] achieve 41.5% $mAP$ and 43.9% $mAP$, which surpasses the baseline by 4.1% and 4.8% respectively, and it also outperforms the previous state-of-the-art methods by a large margin.

In summary, the contributions of this paper are threefold:

- We argue that FPN feature imitation can distill knowledge successfully even though the student-teacher detector pairs are heterogeneous.
- We propose to imitate FPN features with PCC to focus on relational information and relax the distribution constraints of the student feature's magnitude. It is capable of distilling knowledge for both homogeneous and heterogeneous detector pairs.
- We verify the effectiveness of our method on various detectors via extensive experiments on the COCO [25], and achieve state-of-the-art performance without bells or whistles. Moreover, our method converges faster and is not sensitive to the only one hyper-parameter distillation loss weight, which is simple yet effective.

## 2 Related Works

### 2.1 Object Detection

Object detection is considered as one of the most challenging vision tasks which aims at detecting semantic objects of a certain class in images. Modern detectors are roughly divided into two-stage detectors [32, 4, 14] and one-stage detectors [24, 37, 22, 12]. In two-stage detectors, a Region

Proposal Network(RPN) is usually adopted to generate initial rough predictions refined by a task-specific detection head. A typical example is Faster R-CNN [32]. In contrast, One-stage detectors, such as RetinaNet [24] and FCOS [37], can directly and densely predict bounding boxes on the output feature maps. Among these works, multi-scale features are usually adopted to handle objects of various scales, *e.g.* FPN [23], which is considered as a typical case for our study. The proposed PKD only distills the intermediate features and does not rely on detector-specific designs so that it can be used directly on various detectors.

## 2.2 Knowledge Distillation

Knowledge Distillation (KD) is a kind of model compression and acceleration approach aiming at transferring knowledge from a teacher model to a student model. It is popularized by [16] and then its effectiveness in image classification has been explored by subsequent works [33, 17, 15, 1, 46, 29, 27, 36, 38]. However, adapting KD to object detectors is nontrivial since minimizing the Kullback–Leibler (KL) divergence between the classification head outputs fails to transfer the spatial information from the teacher and only brings limited performance gain to the student. The following three strategies are usually adopted by previous methods in detection to handle the above challenge. First, the distillation is usually conducted among multi-scale intermediate features [21], which provides rich spatial information for detection. Second, different feature selection methods are proposed to overcome the foreground-background imbalance. Most of these works can be divided into three kinds according to the feature selection approach [19]: proposal-based [6, 21, 10, 45], rule-based [40, 13] and attention-based [47, 50, 20]. Third, as the relation between different objects contains valuable information, many previous works try to improve the performance of detectors by enabling detectors to capture and make use of these relations, such as non-local modules [47], global distillation [44] and structured instance graph [8].

Unlike the previous works, we consider the magnitude difference, dominant FPN stages and channels as the key problem. We hope our method could serve as a solid baseline and help ease future research in knowledge distillation for object detectors.

## 3 Method

### 3.1 Preliminaries

In this part, we briefly recap the traditional knowledge distillation for object detection. Recently, feature-based distillation over multi-scale features is adopted to deal with rich spatial information for detection. Different imitation masks $M$ are proposed to form an attention mechanism for foreground features and filter away noises in the background. The objective can be formulated as:

$$\mathcal{L}_{FPN} = \sum_{l=1}^{L} \frac{1}{N_l} \sum_{c}^{C} \sum_{i}^{W} \sum_{j}^{H} M_{lcij} \left( F_{lcij}^t - \phi_{adapt} \left( F_{lcij}^s \right) \right)^2, \tag{1}$$

where $L$ is the number of FPN layers, $l$ represents the $l$-th FPN layer, $i$, $j$ are the location of the corresponding feature map with width $W$ and height $H$. $N_l = \sum_{c}^{C} \sum_{i}^{W} \sum_{j}^{H} M_{lcij}$. $F_l^t$ and $F_l^s$ are the $l$-th layer of feature of student and teacher detectors, respectively. Function $\phi_{adapt}$ is a 1x1 convolution layer to upsample the number of channels for the student network if the number of channels mismatches between the teacher and the student.

The definition of $M$ is different in these methods. For example, FRS [50] uses the aggregated classification score map from the corresponding FPN layer, and FGD [44] considers spatial attention, channel attention, object size and foreground-background information simultaneously.

### 3.2 Is FPN feature imitation applicable for heterogeneous detector pairs?

Most of the previous works perform distillation on FPN, as FPN integrates multiple layers of the backbone and provides rich spatial information of multi-scale objects. It is reasonable to force the student to imitate FPN features from its homogeneous teacher as they have the same detection head and label assignment, and better features could lead to better performance. However, there is still a lack of study on how to distill between heterogeneous detectors. [44, 50] conduct experiments

on detectors with heterogeneous backbones, but detectors with heterogeneous detection heads and different label assignments are always omitted. Thus, we are motivated to investigate whether FPN feature imitation still makes sense for these heterogeneous detector pairs.

We conduct backbone and neck replacement experiments on three popular detectors: GFL [22], FCOS [37] and RetinaNet [24]. First, we replace the backbone and neck of FCOS with the well-trained (by 12 epochs) ones of GFL. Since the main idea of feature-based distillation methods is to directly align the feature activations of the teacher and the student, it can be considered as the extreme case of FPN feature imitation between FCOS and GFL. Then the FCOS head is finetuned with the frozen replaced GFL backbone and neck. In Table 1, it can be seen that by replacing with the backbone and neck of GFL, the detector

Table 1: Results of the backbone and neck replacement on COCO.

| Backbone & Neck | Head | $mAP$ |
|---|---|---|
| FCOS-Res50 | FCOS | 36.5 |
| GFL-Res50 | FCOS | 37.6 |
| Retina-Res50 | Retina | 36.3 |
| FCOS-Res50 | Retina | 35.2 |

achieves higher performance (from 36.5 to 37.6). It is verified in some extent that FPN feature imitation is applicable between heterogeneous detectors. In contrast, we replace the backbone and neck of RetinaNet with the well-trained (by 12 epochs) ones of FCOS. Due to feature value magnitude difference between the two models caused by group normalization in FCOS head, a significant mAP drop (from 36.3 to 35.2) can be observed. This means the feature value magnitude difference could interfere with the knowledge distillation between two heterogenous detectors.

### 3.3 Feature Imitation with Pearson Correlation Coefficient

As discussed in Section 3.2, a promising feature distillation approach needs to consider the magnitude difference when constructing them into pairs for imitation. Moreover, by comparing the activation patterns shown in Figure 1, we find the dominant FPN stages and channels can negatively interfere with the training phase of the student and lead to sub-optimal results, which is ignored by previous works.

To address the above issues, we propose first normalizing the features of the teacher and the student to have zero mean and unit variances and minimizing the MSE between normalized features. Additionally, we want the normalization to obey the convolutional property - so that different elements of the same feature map, at different locations, are normalized in the same way. Let $\mathbb{B}$ be the set of all values in a feature map across both the elements of a mini-batch and spatial locations. So for a mini-batch of size $b$ and feature maps of size $h \times w$, we use the effective mini-batch of size $m = \|\mathbb{B}\| = b \cdot hw$. Let $\boldsymbol{s}^{(c)} \in \mathbb{R}^m$ be the $c$-th channel of a batch of FPN outputs and omit $c$ for clarity. Then we get the normalized values $\hat{s}_{1...m}$ and $\hat{t}_{1...m}$ from the student and the teacher, respectively. Instead of delicately designing imitation masks $M$ in Eq. 1 to choose the important features, our PKD operates on the full feature map. That is, the imitation mask $M$ is filled with the scalar value 1. Hence we can formulate our distillation loss as the following:

$$\mathcal{L}_{FPN} = \frac{1}{2m} \sum_{i=1}^{m} (\hat{s}_i - \hat{t}_i)^2. \tag{2}$$

Moving forward, minimizing the loss function above is equivalent to maximizing the PCC between the pre-normalized features of the student and the teacher. PCC can be computed as:

$$r(\boldsymbol{s}, \boldsymbol{t}) = \frac{\sum_{i=1}^{m}(s_i - \mu_s)(t_i - \mu_t)}{\sqrt{\sum_{i=1}^{m}(s_i - \mu_s)^2}\sqrt{\sum_{i=1}^{m}(t_i - \mu_t)^2}}. \tag{3}$$

Since $\hat{\boldsymbol{s}}, \hat{\boldsymbol{t}} \sim \mathcal{N}(0, 1)$, we get $\frac{1}{m-1}\sum_i \hat{s}_i^2 = 1$, $\frac{1}{m-1}\sum_i \hat{t}_i^2 = 1$. Then, we can reformulate Eq. 2 as:

$$\begin{aligned} \mathcal{L}_{FPN} &= \frac{1}{2m}\left((2m - 2) - 2\sum_{i=1}^{m} \hat{s}_i \hat{t}_i\right) \\ &= \frac{2m-2}{2m}(1 - r(\boldsymbol{s}, \boldsymbol{t})) \approx 1 - r(\boldsymbol{s}, \boldsymbol{t}). \end{aligned} \tag{4}$$

The Pearson correlation coefficient is essentially a normalized measurement of the covariance, such that the result always has a value between $-1$ and $1$. Hence, $L_{FPN} = 1 - r$ always has a value

between 0 and 2. It focuses on the linear correlation between the features of the teacher and the student, and relaxes constraints on the magnitude of the features. Actually, the feature maps $s, t \in \mathbb{R}^m$ can be regarded as $m$ data points $(s_i, t_i)$. $L_{FPN} = 0$ implies that all data points lie on a line for which $s$ increases as $t$ increases. Hence the student is well-trained. And vice versa for $L_{FPN} = 2$. A value of 1 implies that there is no linear dependency between the features of the student and the teacher.

During training, the gradient of PCC, $\partial \mathcal{L}/\partial s_i$, with respect to each FPN output $s_i$, is given by:

$$\frac{\partial \mathcal{L}_{FPN}}{\partial s_i} = \frac{1}{m\sigma_s}(\hat{s}_i \cdot r(\boldsymbol{s}, \boldsymbol{t}) - \hat{t}_i), \tag{5}$$

where $r(\boldsymbol{s}, \boldsymbol{t})$ is PCC in Eq. 3 and $\sigma_s$ is the sample standard deviation of the student's feature $\boldsymbol{s}$.

In conclusion, PCC focuses on the relational information from the teacher and relaxes the distribution constraints on the student feature's magnitude. Moreover, it eliminates the negative impacts of the dominant FPN stages and channels, leading to better performance. As a result, the normalization mechanism bridge the gap between the activation patterns of the student and the teacher (see Figure 2). Hence, feature imitation with PCC works for heterogeneous detector pairs. We train the student detector with the total loss as follows:

$$\mathcal{L} = \mathcal{L}_{GT} + \alpha \mathcal{L}_{FPN}, \tag{6}$$

where $\mathcal{L}_{GT}$ is the detection training loss, $\alpha$ is the hyper-parameter to balance the detection training loss and distillation loss.

## 3.4 Connection of PCC and KL divergence

As discussed in Section 3.3, the normalization mechanism is the key to addressing the three issues mentioned above. In previous works [16, 41, 34, 49], KL divergence has been widely used in distillation. They first convert activations into a probability distribution with softmax function and then minimize the asymmetry KL divergence of the normalized activation maps:

$$\mathcal{L}_{KL} = T^2 \sum_{i=1}^{m} \phi(t_i) \cdot log[\frac{\phi(t_i)}{\phi(s_i)}], \tag{7}$$

where $T$ is a hyper-parameter to control the degree of softness of the targets, and $\phi$ is the softmax function:

$$\phi(t) = \frac{exp(t_i/T)}{\sum_{j=1}^{m} exp(t_j/T)}. \tag{8}$$

Here, we show that minimizing KL divergence between normalized features in the high-temperature limit is equivalent to minimizing MSE between normalized ones, and hence equivalent to maximizing PCC between original ones.

Let $p_i = \phi(\hat{t}_i)$ and $q_i = \phi(\hat{s}_i)$ denote the probabilities from the teacher and the student, respectively. The KL divergence gradient, $\partial \mathcal{L}_{KL}/\partial \hat{s}_i$, with respect to each normalized activation $\hat{s}_i$ of the student is given by:

$$\frac{\partial \mathcal{L}_{KL}}{\partial \hat{s}_i} = T(q_i - p_i) = T\left(\frac{e^{\hat{s}_i/T}}{\sum_j e^{\hat{s}_j/T}} - \frac{e^{\hat{t}_i/T}}{\sum_j e^{\hat{t}_j/T}}\right). \tag{9}$$

If $T$ is large compared with the magnitude of the normalized activations, we can approximate:

$$\frac{\partial \mathcal{L}_{KL}}{\partial \hat{s}_i} \approx T\left(\frac{1 + \hat{s}_i/T}{m + \sum_j \hat{s}_j/T} - \frac{1 + \hat{t}_i/T}{m + \sum_j \hat{t}_j/T}\right). \tag{10}$$

As $\hat{s}_i$ and $\hat{t}_i$ have been zero-meaned, Eq. 10 simplifies to:

$$\frac{\partial \mathcal{L}_{KL}}{\partial \hat{s}_i} \approx \frac{1}{m}\left(\hat{s}_i - \hat{t}_i\right). \tag{11}$$

Employing the relation between MSE and PCC shown in Eq. 4 gives the desired result. Experimental results are provided in Section A.3.

Table 2: Results of the proposed method with different detection frameworks on the COCO dataset. T and S mean the teacher and student detector, respectively. * indicates the results reproduced by us.

| Method | schedule | $mAP$ | $AP_{50}$ | $AP_{75}$ | $AP_S$ | $AP_M$ | $AP_L$ |
|---|---|---|---|---|---|---|---|
| Retina-ResX101 (T) | 2x | 40.8 | 60.5 | 43.7 | 22.9 | 44.5 | 54.6 |
| Retina-Res50 (S) | 2x | 37.4 | 56.7 | 39.6 | 20.0 | 40.7 | 49.7 |
| FKD [47] | 2x | 39.6 (+2.2) | 58.8 | 42.1 | 22.7 | 43.3 | 52.5 |
| FRS [50] | 2x | 40.1 (+2.7) | 59.5 | 42.5 | 21.9 | 43.7 | 54.3 |
| FGD [44] | 2x | 40.4 (+3.0) | 59.9 | 43.3 | **23.4** | 44.7 | 54.1 |
| PKD (Ours) | 2x | **40.8 (+3.4)** | **60.3** | **43.4** | 23.0 | **45.1** | **54.7** |
| FasterRCNN-Res101 (T) | 2x | 39.8 | 60.1 | 43.3 | 22.5 | 43.6 | 52.8 |
| FasterRCNN-Res50 (S) | 2x | 38.4 | 59.0 | 42.0 | 21.5 | 42.1 | 50.3 |
| GID [10] | 2x | 40.2 (+1.8) | 60.7 | 43.8 | 22.7 | 44.0 | 53.2 |
| FRS [50] | 2x | 40.4 (+2.0) | 60.8 | 44.0 | **23.2** | 44.4 | 53.1 |
| FGD [44] | 2x | 40.4 (+2.0) | 60.7 | 44.3 | 22.8 | 44.5 | **53.5** |
| PKD (Ours) | 2x | **40.5 (+2.1)** | **60.9** | **44.4** | 22.6 | **44.8** | 53.1 |
| FCOS-Res101 (T) | 2x+ms | 41.2 | 60.4 | 44.2 | 24.7 | 45.3 | 52.7 |
| FCOS-Res50 (S) | 2x+ms | 39.1 | 58.4 | 41.6 | 24.0 | 42.7 | 48.7 |
| FRS [50] * | 2x+ms | 42.2 (+3.1) | 60.6 | 45.6 | **27.1** | 46.5 | 53.0 |
| FGD [44] * | 2x+ms | 42.3 (+3.2) | 60.8 | 45.8 | 26.1 | 46.7 | 53.3 |
| PKD (Ours) | 2x+ms | **42.8 (+3.7)** | **61.4** | **46.2** | 25.9 | **47.2** | **54.6** |
| RepPoints ResNeXt101 (T) | 2x | 44.2 | 65.5 | 47.8 | 26.2 | 48.4 | 58.5 |
| RepPoints Res50 (S) | 2x | 38.6 | 59.6 | 41.6 | 22.5 | 42.2 | 50.4 |
| FGD [44] | 2x | 41.3 (+2.7) | - | - | **24.5** | 45.2 | 54.0 |
| PKD (Ours) | 2x | **42.3 (+3.7)** | **63.1** | **45.4** | 23.9 | **46.6** | **56.5** |
| TOOD-ResX101 (T) | 2x+ms | 47.6 | 68.5 | 51.6 | 30.6 | 51.4 | 59.7 |
| TOOD-Res50 (S) | 1x | 42.4 | 59.7 | 46.2 | 25.4 | 45.5 | 55.7 |
| PKD (Ours) | 1x | **45.5 (+3.1)** | **62.8** | **49.3** | **27.4** | **49.8** | **58.4** |

## 4 Experiments

In order to verify the effectiveness and robustness of our method, we conduct experiments on different detection frameworks on the COCO [25] dataset. We choose the default 120k images split for training and 5k images split for the test. We use the standard training settings following [44] and report mean Average Precision ($AP$) as an evaluation metric, together with $AP$ under different thresholds and scales, $i.e.$, $AP_{50}$, $AP_{75}$, $AP_S$, $AP_M$ and $AP_L$.

For distillation, the hyper-parameter $\alpha$ is set to 6 when using a two-stage detector as the teacher and 10 when using a one-stage one. Feature levels of heterogeneous detector pairs may not be strictly aligned, $e.g.$, FasterRCNN constructs the feature pyramid from $P2$ to $P6$, while RetinaNet uses $P3$ to $P7$. To address the above issue, we upsample the low-resolution feature maps to have the same spatial size as the high-resolution ones. In addition, the adaptive layer ($\phi_{adapt}$ in Eq. 1) is unnecessary for our PKD.

All experiments are performed on 8 Tesla A100 GPUs with two images in each. Our implementation is based on mmdetection [7] and mmrazor [9] with Pytorch framework [30]. '1x' (namely 12epochs), '2x' (namely 24 epochs) and '2x+ms' (namely 24epochs with multi-scale training) training schedules are used. More details are given in the Appendix.

### 4.1 Main Results

Our method can be applied to different detection frameworks easily, so we first conduct experiments on five popular detectors, including a two-stage detector (Faster RCNN[32]), two anchor-based one-stage detector (RetinaNet [24], RepPoints [43] and TOOD [12]) and an anchor-free detector (FCOS [37]). Table 2 shows the comparison of the results of state-of-the-art distillation methods on the COCO. Our distillation method surpasses other state-of-the-art methods. All the student detectors

gain significant improvements in $AP$ with the knowledge transferred from teacher detectors, *e.g.*, FCOS with ResNet-50 gets a 3.7% $mAP$ improvement on the COCO dataset. These results indicate the effectiveness and generality of our method in both one-stage and two-stage detectors.

### 4.2 Distilling More Student Detectors with Stronger Heterogeneous Teachers

Most of the current methods are designed for homogeneous detector pairs. PKD is general enough to distill knowledge between both homogeneous and heterogeneous detector pairs. Here we conduct experiments on more detectors and use stronger heterogeneous teacher detectors, as shown in Table 3. Comparing with Table 2, we find that student detectors perform better with stronger teacher detectors, *e.g.*, Retina-Res50 model achieves 41.5% and 39.6% $mAP$ with Mask RCNN-Swin [28] and Retina-Res101, respectively. Results show that mimicking the feature maps of stronger heterogeneous teacher detectors can further boost the student's performance when applied with our PKD.

Table 3: Results of more detectors with stronger heterogeneous teacher detectors on the COCO dataset.

| Method | schedule | $mAP$ | $AP_{50}$ | $AP_{75}$ | $AP_S$ | $AP_M$ | $AP_L$ |
|---|---|---|---|---|---|---|---|
| Mask RCNN-Swin (T) | 3x+ms | 48.2 | 69.8 | 52.8 | 32.1 | 51.8 | 62.7 |
| Retina-Res50 (S) | 2x | 37.4 | 56.7 | 39.6 | 20.0 | 40.7 | 49.7 |
| PKD (Ours) | 2x | **41.5 (+4.1)** | **60.6** | **44.1** | **22.9** | **45.2** | **56.4** |
| Mask RCNN-Swin (T) | 3x+ms | 48.2 | 69.8 | 52.8 | 32.1 | 51.8 | 62.7 |
| FCOS-Res50 (S) | 2x+ms | 39.1 | 58.4 | 41.6 | 24.0 | 42.7 | 48.7 |
| PKD (Ours) | 2x+ms | **43.9 (+4.8)** | **62.3** | **47.5** | **27.2** | **48.0** | **57.1** |
| GFL-Res101 (T) | 2x+ms | 44.9 | 63.1 | 49.0 | 28.0 | 49.1 | 57.2 |
| FCOS-Res50 (S) | 2x+ms | 39.1 | 58.4 | 41.6 | 24.0 | 42.7 | 48.7 |
| PKD (Ours) | 2x+ms | **43.5 (+4.4)** | **61.9** | **47.1** | **26.5** | **47.7** | **55.7** |

Table 4: Results of other feature-based distillation methods with the normalized feature maps.

| FitNet | GT Mask | FRS feature imitation | FGD Global | PKD | GFL101 (44.7) GFL50 (40.2) |
|---|---|---|---|---|---|
| ✓ | | | | | 41.3 |
| ✓ | | | | ✓ | 43.3 |
| | ✓ | | | | 41.4 |
| | ✓ | | | ✓ | 43.0 |
| | | ✓ | | | 43.2 |
| | ✓ | | | ✓ | 43.4 |
| | | | ✓ | | 40.2 |
| | | | ✓ | ✓ | 43.3 |

### 4.3 Other feature-based distillation methods with the normalized features

In object detection, some feature-based distillation methods [50, 44] transfer the knowledge within some pre-defined imitation regions, and achieved competitive results. In contrast, we believe full map feature imitation with our PKD can already outperform them. In order to show the generality of our method, we take FitNet [33], GT Mask, FRS [50] and FGD [44] as baselines and build PKD on them.

**Recaps.** Full feature map distillation with MSE loss is used in FitNet. While the GT Mask only imitates features that overlap with ground truth bounding boxes. Feature imitation with FRS uses the aggregated classification score, takes maximum operation in channel direction, of the classification head output as a weighted score mask, to guide the distillation of the FPN. And global distillation in FGD rebuilds the relation between different pixels and transfers it from teachers to students through GcBlock [5].

**Results.** For a fair comparison, all the hyper-parameters are copied from the original paper. Results in Table 4 show that combining PKD with these feature imitation methods can further improve their performance.

Table 5: Comparisons between feature imitation with MSE and our PKD. We address the above three issues and achieve better performance.

| Detector Pairs | Methods | $mAP$ | $AP_S$ | $AP_L$ |
|---|---|---|---|---|
| FCOS-ResX101 Retina-Res50 | MSE | 36.3 | 20.0 | 47.1 |
| | Ours | 41.3 | 24.2 | 55.4 |
| GFL-Res101 FCOS-Res50 | MSE | 43.0 | 27.1 | 54.3 |
| | Ours | 43.5 | 26.5 | 55.7 |
| Retina-ResX101 Retina-Res50 | MSE | 40.4 | 22.2 | 54.4 |
| | Ours | 40.8 | 23.0 | 54.7 |

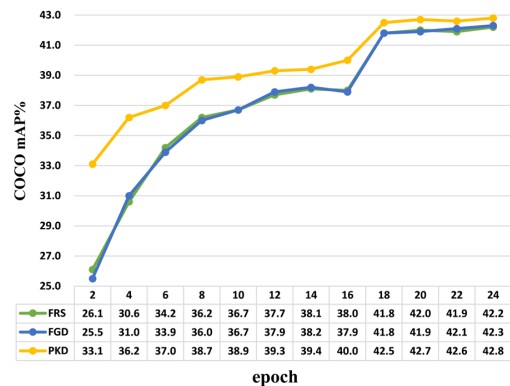

| | 2 | 4 | 6 | 8 | 10 | 12 | 14 | 16 | 18 | 20 | 22 | 24 |
|---|---|---|---|---|---|---|---|---|---|---|---|---|
| FRS | 26.1 | 30.6 | 34.2 | 36.2 | 36.7 | 37.7 | 38.1 | 38.0 | 41.8 | 42.0 | 41.9 | 42.2 |
| FGD | 25.5 | 31.0 | 33.9 | 36.0 | 36.7 | 37.9 | 38.2 | 37.9 | 41.8 | 41.9 | 42.1 | 42.3 |
| PKD | 33.1 | 36.2 | 37.0 | 38.7 | 38.9 | 39.3 | 39.4 | 40.0 | 42.5 | 42.7 | 42.6 | 42.8 |

Figure 3: Comparison of the convergence speed of FCOS-Res50 among other state-of-the-art distillation methods and ours.

## 4.4 Analysis

### 4.4.1 Effectiveness of Pearson Correlation Coefficient

Among most previous works, feature-based distillation over multi-scale features is adopted for distillation. In this study, we argue that the magnitude difference, dominant FPN stages and channels can negatively interfere with the training phase of the student and lead to sub-optimal results. To study this empirically, we conduct the following three experiments to explore the effects of MSE loss which is a widely-used loss function in distillation and suffers from the above three issues simultaneously. For a fair comparison, we tune the loss weight of MSE for all experiments. More details are listed in Section A.4.

For the first pair, FCOS-ResX101 is used as the teacher whose feature magnitude is significantly different from that of the student Retina-Res50 (refer to Figure 4). In this case, what constitutes the knowledge is better presented by relational information from the teacher's features than the absolute values. For the second pair, GFL-Res101 is the teacher and FCOS-Res50 is the student. For both teacher and student, features in FPN stages 'P5' and 'P6' are less activated than those in stages 'P3' and 'P4' (refer to Figure 5). Hence, features in stages 'P3' and 'P4' could dominate the gradient of the distillation loss, which will overwhelm the effects of other features. Since larger objects are usually assigned to higher feature levels, the student's performance for large objects is significantly lower than that of ours when MSE is used as the distillation loss. For the last pair, Retina-ResX101 is the teacher and Retina-Res50 is the student. There are always a few channels with greater values (refer to Figure 6 and Figure 7) , so directly imitating the feature maps may introduce much noise in dominant channels.

By comparing the results in Table 5, we find that our proposed PKD addresses the above three issues and achieves better performance. Hence, an effective distillation loss function should have the ability to handle the above three problems. We hope PKD could serve as a solid baseline and help ease future research in knowledge distillation community.

### 4.4.2 Convergence Speed

In this subsection, we conduct experiments with FCOS to compare the convergence speed of our method with other state-of-the-art distillation methods on the COCO benchmark. As shown in Figure 3, the training convergence can significantly speed up in the early training stage with our PKD. Meanwhile, the final performance of the student detector FCOS-Res50 is about 0.5%-0.6% $mAP$ higher than that of FGD and FRS. In addition, there is no need to forward the detection head of the

teacher, which reduces the training time significantly, especially for those models with cascaded heads.

### 4.4.3 Sensitivity study of loss weight $\alpha$

In Eq. 6, we use the loss weight hyper-parameter $\alpha$ to balance the detection training loss and distillation loss. Here, we conduct several experiments to investigate the influence of $\alpha$. As shown in Table 6, the worst result is just a 0.3 $mAP$ drop compared with the best result, indicating our method is not sensitive to the only hyper-parameter $\alpha$. In addition, thanks to the normalization mechanism in PCC, a relatively uniform value of loss weight $\alpha$ can be found among different teacher-student detector pairs to keep the balance of the detection loss and the distillation loss unchanged.

Table 6: Ablation study of loss weight hyper-parameter $\alpha$ on FCOS ResX101 - RetinaNet Res50.

| $\alpha$ | 3 | 5 | 8 | 10 | 13 |
|---|---|---|---|---|---|
| $mAP$ | 41.0 | 41.1 | 41.1 | 41.3 | 41.1 |

## 5 Conclusion

This paper empirically finds that FPN feature imitation is applicable for heterogeneous detector pairs although their detection heads and label assignments are different. Then we propose feature imitation with Pearson Correlation Coefficient to focus on the relational information from the teacher and relax the distribution constraints of the student's feature value magnitude. Furthermore, a general KD framework is proposed, capable of distilling knowledge for both homogeneous and heterogeneous detector pairs. It converges faster and only introduces one hyper-parameter, which can easily be applied to other datasets. However, our understanding of whether our proposed PKD is capable of other tasks such as text recognition is preliminary and left as future works.

## Acknowledgments

This work is supported in part by the Shanghai Committee of Science and Technology, China (Grant No. 20DZ1100800), NSFC 62273347, the National Key Research and Development Program of China (2020AAA0103402), Jiangsu Key Research and Development Plan (No.BE2021012-2) and NSFC 61876182.

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
