# A Appendix

## A.1 Motivation to build a general KD framework

Modern detectors are roughly divided into two-stage detectors [32, 14, 3] and dense prediction detectors (*e.g.*, anchor-based one-stage detectors [24, 26, 31] and anchor-free one-stage detectors [11, 37, 43]). Each family has its own advantages and weakness. In particular, two-stage detectors usually have higher performance, while being slower in inference speed and harder to be deployed due to the region proposal network (RPN) and RCNN head. On the other hand, dense prediction detectors are faster than two-stage detectors while being less accurate. In practice, it is a natural idea to use two-stage detectors as teachers to enhance dense prediction detectors.

Moreover, knowledge distillation between heterogeneous dense prediction detector pairs is also promising. In some scenarios, only the detectors with a specific architecture can be deployed due to hardware limitations. For example, compared with Batch Normalization [18] and Instance Normalization [39], Group Normalization [42] is hard to deploy. However, the most powerful teachers may belong to different categories. Furthermore, object detection is developing rapidly and algorithms with better performance are proposed continuously. Nevertheless, it is not easy to change detectors frequently in terms of stability and hardware runtime limitations in practical applications. So it is helpful if knowledge distillation can be conducted between the latest high-capacity detectors and the widely-used compact detectors.

Hence, we are motivated to design a general distillation method capable of distilling knowledge between both homogeneous and heterogeneous detector pairs.

## A.2 Details of Training Recipe

We conduct experiments on different detection frameworks, including two-stage models, anchor-based one-stage models and anchor-free one-stage models. [19] proposes inheriting strategy which initializes the student with the teacher's neck and head parameters and gets better results. Here we use this strategy to initialize the student which has the same head structure as the teacher and find that it helps students converge faster.

All experiments are performed on 8 Tesla A100 GPUs with 2 images in each. Our implementation is based on mmdetection [7] and mmrazor [9] with Pytorch framework [30]. '1x' (namely 12 epochs), '2x' (namely 24 epochs) and '2x+ms' (namely 24 epochs with multi-scale training) training schedules with SGD optimizer are used. Momentum and weight decay are set to 0.9 and 1e-4. The initial learning rate is set to 0.02 for Faster RCNN and 0.01 for others. We train FCOS [37] with tricks including GIoULoss, norm-on-bbox and center-sampling which is the same as FGD [44] and GID [10]. For distillation, only one hyper-parameter $\alpha$ is introduced to balance the supervised learning loss and distillation loss, and it is set to 6 when using a two-stage detector as the teacher and 10 when using a one-stage one as the teacher. The teacher network is well-trained previously and fixed during training.

## A.3 Connection of PCC and KL divergence

Section 3.4 in the main text shows the connection between PCC and KL divergence. We conduct two experiments on RetinaNet and GFL to verify this empirically. For both experiments, we set loss weight $\alpha = 10$ and temperature $T = 50$. As shown in Table 7, minimizing KL divergence between post-normalized features in the high-temperature limit can also achieve similar results.

## A.4 Details of feature imitation with MSE

As shown in Table 5 in the main text, we compare the results of MSE with our proposed PKD. As the gradient of MSE loss depends on feature value magnitude and it is usually different among different detectors, we have to tune the loss weight carefully to achieve relatively good results, as shown in Table 8. And we put the best results down in Table 5.

Table 7: Results of the KL divergence with normalization mechanism.

| Method | schedule | $mAP$ | $AP_{50}$ | $AP_{75}$ | $AP_S$ | $AP_M$ | $AP_L$ |
|---|---|---|---|---|---|---|---|
| Retina-ResX101 (T) | 2x | 40.8 | 60.5 | 43.7 | 22.9 | 44.5 | 54.6 |
| Retina-Res50 (S) | 2x | 37.4 | 56.7 | 39.6 | 20.0 | 40.7 | 49.7 |
| Norm+KL | 2x | 40.9 | 60.3 | 43.6 | 22.9 | 45.2 | 55.1 |
| GFL-Res101 (T) | 2x+ms | 44.9 | 63.1 | 49.0 | 28.0 | 49.1 | 57.2 |
| GFL-Res50 (S) | 1x | 40.2 | 58.4 | 43.3 | 23.3 | 44.0 | 52.2 |
| Norm+KL | 1x | 43.1 | 60.9 | 46.7 | 25.1 | 47.8 | 55.9 |

Table 8: Results of MSE on various detector pairs.

| Teacher | Student | Schedule | Baseline | Loss Weight | $mAP$ |
|---|---|---|---|---|---|
| FCOS-ResX101 | Retina-Res50 | 1x | 36.5 | 5 | 33.9 |
| | | | | 10 | 31.4 |
| | | | | 20 | 29.7 |
| FCOS-ResX101 | Retina-Res50 | 2x | 37.4 | 5 | 36.3 |
| | | | | 10 | 35.6 |
| | | | | 20 | 34.9 |
| GFL-Res101 | FCOS-Res50 | 1x | 36.6 | 50 | 38.7 |
| | | | | 70 | 39.2 |
| GFL-Res101 | FCOS-Res50 | 2x | 39.1 | 10 | 41.5 |
| | | | | 30 | 42.5 |
| | | | | 50 | 42.7 |
| | | | | 70 | 43.0 |
| | | | | 80 | 42.9 |
| Retina-ResX101 | Retina-Res50 | 2x | 37.4 | 5 | 40.0 |
| | | | | 10 | 40.4 |
| | | | | 15 | 40.3 |
| MaskRCNN-Swin | FasterRCNN-Res50 | 2x | 38.4 | 3 | 41.7 |
| | | | | 5 | 41.7 |
| | | | | 6 | 41.6 |
| | | | | 10 | 41.7 |

## A.5  Effectiveness of Pearson Correlation Coefficient

In this paper, we argue that the magnitude difference, dominant FPN stages and channels can negatively interfere with the training phase of the student. To clearly show that these three issues do exist, we elaborately visualize the FPN feature responses of the teacher and student detectors before distillation, as shown in Figure 4, Figure 5, Figure 6 and Figure 7. We follow the visualization method in Section 1.

Through these comparisons, we obtain the following three observations:

(1) The feature value magnitude of different detectors could be significantly different, especially for heterogeneous detectors, as shown in Figure 4. Directly aligning the feature maps between the teacher and the student may enforce overly strict constraints and do harm to the student (see Table 5 top in the main text).

(2) As shown in Figure 5, compared with features in FPN stage 'P3' and 'P4', features in stage 'P5' and 'P6' are less activated in both GFL-Res101 and FCOS-Res50. And it is a common case among different detector pairs. FPN stages with larger values could dominate the gradient of the distillation loss, which will overwhelm the effects of other features in KD.

(3) As shown in Figure 6 (right), the feature magnitude of the 210-th channel of MaskRCNN-Swin FPN stage 'P6' is significantly larger than others. Similar phenomena also exist in other detectors such as RetinaNet (see Figure 7). The small gradients produced by those less activated channels can be drowned in the large gradients produced by dominant ones, thus limiting further refinement. Furthermore, there is much noise in the object-irrelevant area, as depicted in Figure 6 (left). Directly imitating the feature maps may introduce much noise.

Comparing Table 8 and Table 5 in the main text, we find that our proposed PKD addresses the above three issues and achieves better performance. Hence, an effective distillation loss function should have the ability to handle the above three problems. We hope PKD could serve as a solid baseline and help ease future research in knowledge distillation community.

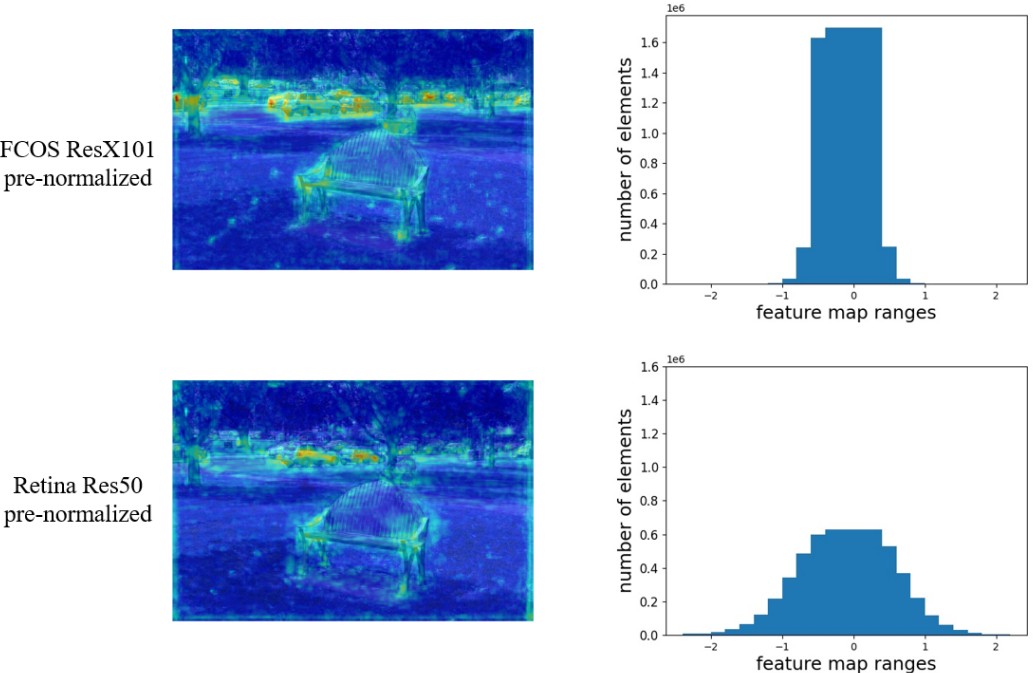

Figure 4: Visualization of the activation patterns and activation distribution of FPN stage 'P3'.

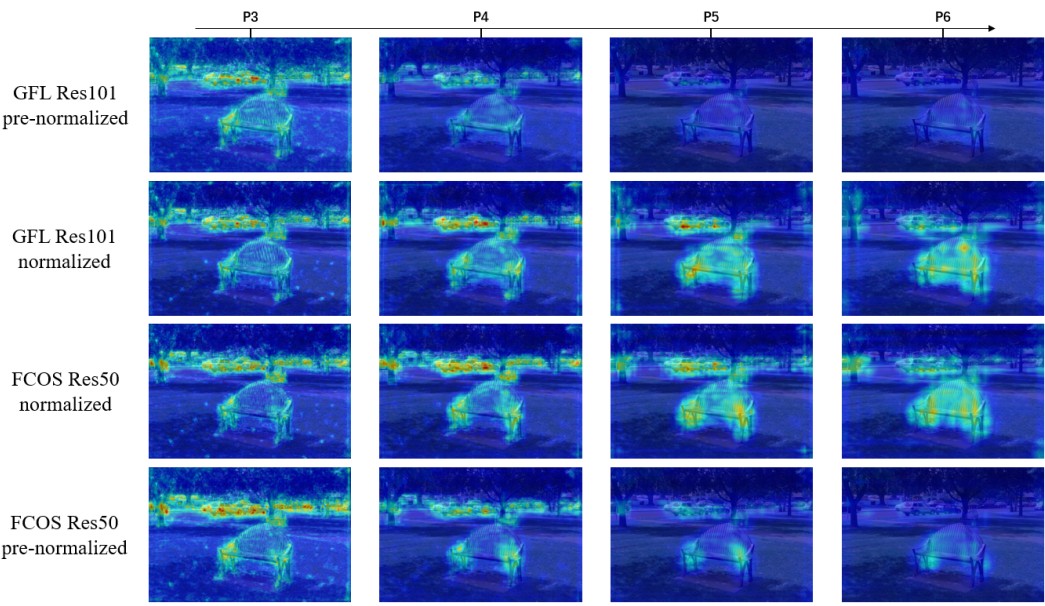

Figure 5: Visualization of dominant FPN stages. From left to right, they correspond to the activation patterns in FPN stage 'P3' to 'P6'. The leftmost corresponds to the lowest stage of FPN, and the rightmost corresponds to the highest stage of FPN.

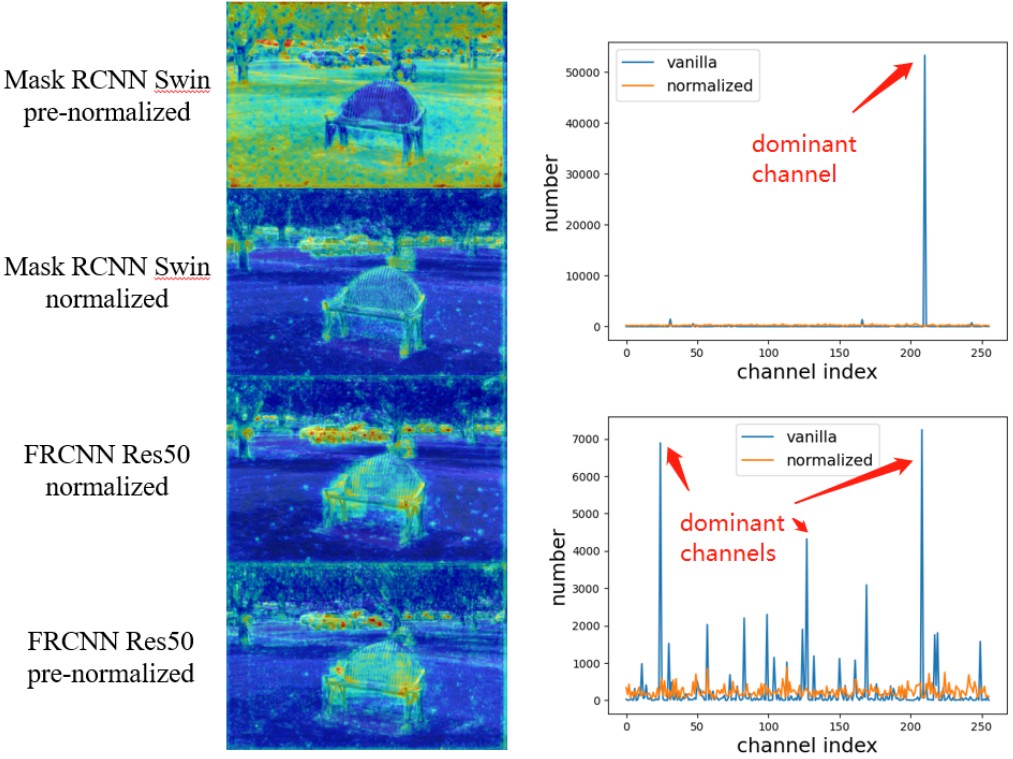

Figure 6: Visualization of dominant channels in MaskRCNN-Swin and FasterRCNN-Res50. Left: Visualization of the activation patterns of FPN stage 'P3'. Right: Dominant channels in pre-normalized FPN stage 'P3'. Let $s_{l,u,v} \in \mathbb{R}^C$ denote the feature vector located in pixel (u, v) from $l$-th FPN stage and omit $l$ for clarity. Then $number_i = \sum_{u,v} \mathbb{1}[\arg\max_c s_{u,v}^{(c)} = i]$ where $i$ is the channel index. We define channels with a larger $number$ as dominant channels.

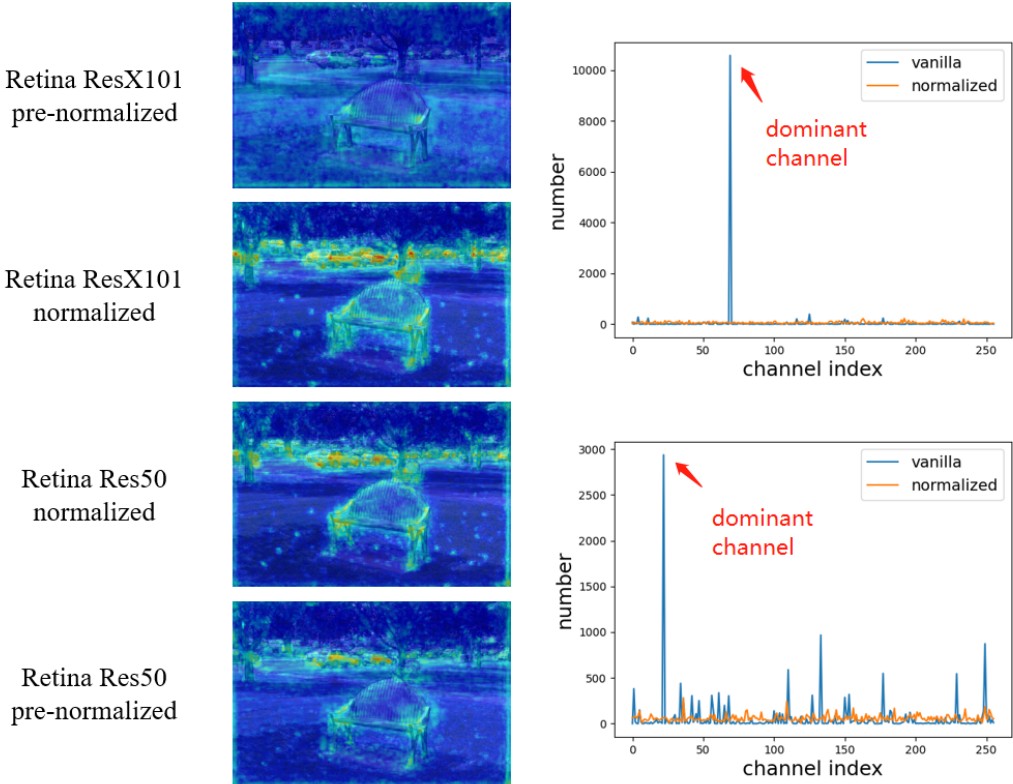

Figure 7: Visualization of dominant channels in Retina-ResX101 and Retina-Res50.

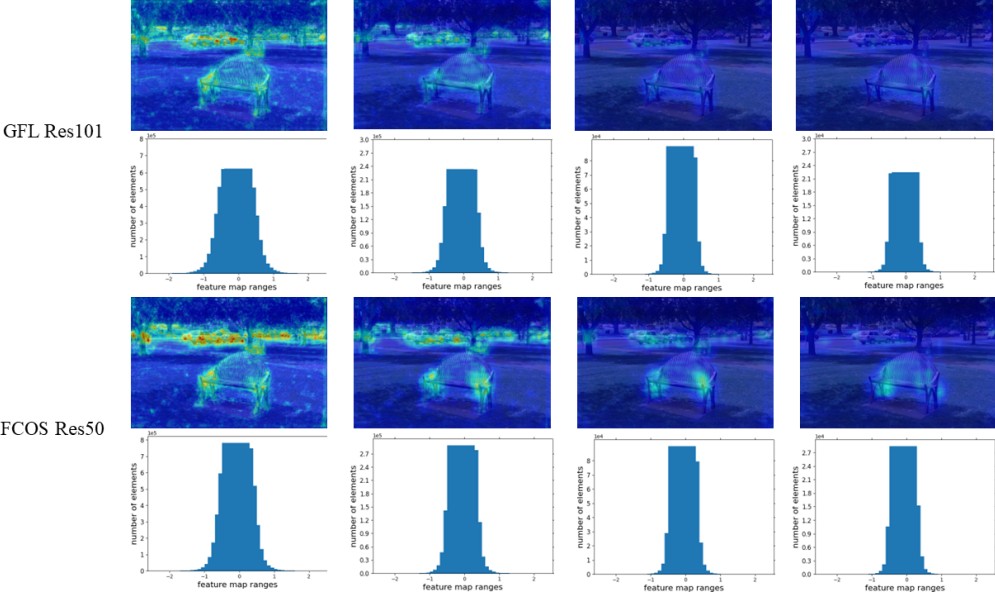

Figure 8: Visualization of the activation patterns and activation distribution of GFL-Res101 and FCOS-Res50. From left to right, they correspond to the activation patterns and activation distribution in FPN stage 'P3' to 'P6'. The leftmost corresponds to the lowest stage of FPN, and the rightmost corresponds to the highest stage of FPN. The feature value magnitude of GFL and FCOS is similar.