# OpenReview forum: "PKD: General Distillation Framework for Object Detectors via Pearson Correlation Coefficient"
_NeurIPS.cc/2022/Conference — NeurIPS 2022 Accept_

### Official Review · Reviewer_JKs3 · 2022-06-20

**Rating:** 6
**Confidence:** 5
**Soundness:** 3 good
**Presentation:** 2 fair
**Contribution:** 3 good

**Summary:**

Feature imitation is one of the widely used knowledge distillation techniques, where MSE loss is widely used to enforce the consistency of feature representation between the teacher-student pair. This paper proposes a new loss function for feature imitation in object detection. It first conducts feature standardization and then calculates the MSE loss. Methematically, it is equivalent to firstly calculate the Pearson Correlation Coefficient $r$ between two feature vectors, and then use 1-$r$ as the new feature imitation loss. Expreiments show that it can be better than traditional MSE loss in distilling both heterogeneous detector pairs and homogeneous detector pairs.

**Questions:**

see weakness

**Limitations:**

I like the novelty, the effectiveness, and the simplicity of this paper. However, my pre-rebuttal rating score is 4 because I have some major concerns about the arguments and experiments.

The most important concerns are Weakness 1, 2, 3. If they are sovled correctly, I will raise my rating score. If there still exists one of the Weakness 1, 2, 3, I will not change the rating score. I'm happy to discuss with the authors and other reviewers and exchange the ideas about this paper.

The story needs to be reorganized. The paper needs a major revision.

The best modifications are:

1. See Weakness 1, remove the fine-tuning Table 1. Combining Fig. 1 and Table B in the Appendix "FCOS-ResX101 $\rightarrow$ Retina-Res50 1x 33.9 (-2.6)" "GFL-Res101 $\rightarrow$ FCOS-Res50 1x 39.2 (+2.6)". Then you make the argument that the feature misalignment, or something like the feature magnitude difference, severely affects the performance of feature imitation. According to these observations, we propose ... We find that feature standardization is the key to effectively mitigate ... As shown in Fig 2, the feature standardization can ...

2. See Weakness 2, 5 and 7.

3. See Weakness 3.

The authors should also describe the property of the proposed loss function, about its boundness, optimization goal (linear correlation of two features), more relaxed than MSE loss, and the limit cases (if $\mathcal{L}_{FPN}\rightarrow0,1,2$, what are the meanings of these situations.)

After making all the above modification, it can meet the acceptance bar.

**Strengths And Weaknesses:**

Strengths:

1. I highly like the contribution of this paper, because I've also observed before that this idea, the correlation imitation loss (replace MSE with 1-PCC) did work. And it did speed up the training process, in other words, it did reduce the training difficulty in the early training stage. However, from my observation, the performance improvement is limited. The AP can be further improved by only +0.2 based on LD [r2] +KD+GI imitation [9] (42.4$\rightarrow$42.6, GFLv1 Res101 2x $\rightarrow$ GFLv1 Res50 1x). I just simply replaced the MSE loss of GI imitation with correlation imitation loss (1-PCC), and searched a best loss weight of course.

2. This paper also incorporate their method to distill heterogeneous detector and demonstrate that PKD works better than traditional MSE loss. They find that feature standardization is the key to conduct feature imitation for distilling the heterogeneous detectors. This is really a good idea.

3. Due to its novelty, effectiveness and simplicity, it has a potential to serve as a strong baseline for future works on knowledge distillation for object detection.

Weaknesses:

1. In Sec. 3.2, **the fine-tuning experiments cannot support your argument** “Hence, FPN feature imitation is applicable for heterogeneous detectors after removing the negative impacts of magnitude difference.” (line 158-159). If you want to make that argument, you must list the results of Table 4 and need further improvement. For example, FCOS-ResX101 $\rightarrow$ Retina-Res50 MSE is worse than baseline (use Table B in the Appendix). GFL-Res101 $\rightarrow$ FCOS-Res50 MSE and Retina-ResX101 $\rightarrow$ Retina-Res50 MSE works well. This phenomenon combined with the feature magnitude difference, **not fine-tuning**, shows the importance of feature standardization which aligns the features between the heterogeneous detectors so as to make the feature imitation more efficient. As for the homogeneous detectors, the feature magnitude difference is relatively small. And your PKD can still produce a slight improvement. (This is very important to the whole story and it affects my rating score by at least 1 point.)

2. From Table 4, it can be seen that PKD can only improve the performance by +0.4~0.5 AP over MSE loss and even no improvement (RetinaNet Res101 $\rightarrow$ Res50) if homogeneous teachers are used. In Table 2, you compare PKD with various KD methods such as FRS and FGD. It is confusing to me that Retina-ResX101 $\rightarrow$ Retina-Res50 FGD gets 40.4 AP and your PKD gets 40.8 AP, while GFL-Res101 $\rightarrow$ GFL-Res50 FGD gets 41.3 AP and PKD gets 43.3 AP. **Why is there such a big difference?** I notice that you implement your method by setting the imitation mask to 1, which means that your PKD is based on FitNet, i.e., full map feature imitation, right? If it is true, your results in Table 2 is weird to me, e.g., GFL-Res101 $\rightarrow$ GFL-Res50 1x PKD is 2 point higher than FGD, but 0.3 lower than FRS. It seems that PKD+FitNet can already outperform these feature imitation methods, like FRS and FGD. However, I believe that PKD can be combined with these feature imitation methods since only the MSE loss needs to be replaced. And this is exactly what you show in Table 4. Therefore, you must check the context in Sec. 3.3 and the experiments, and **express the implementation details correctly**. You must emphasize which feature imitation method the PKD results in Table 2 are based on.

3. In 4.3.3 Convergence Speed, the comparison of Table 5 is **unfair**. You choose FCOS-ResX101 as the teacher, while all the others are Retina-Res101. It must be the same Retina-Res101 to compare your PKD with the others. Notice that, in Table 6, FCOS ResX101 $\rightarrow$ RetinaNet Res50 2x gets 41.3 AP, i.e., the 2x training schedule can get higher AP than 1x, which is about +1.0. And in Table 4, Retina-ResX101 $\rightarrow$ Retina-Res50 PKD 2x gets 40.8 AP. Therefore, the performance of Retina-Res101 $\rightarrow$ Retina-Res50 PKD 1x seems to be 39.3 around. BTW, from my observation, the training convergence can be significantly speed up in the early training stage. Here are my previous results: GFLv1 Res101 2x $\rightarrow$ GFLv1 Res50 1x.

|Method / epoch | 1 | 2 | 3 | 4 | 5 | 6 | 7 | 8 | 9 | 10 | 11 | 12 |
|:--------:|:--------:|:--------:|:--------:|:--------:|:--------:|:--------:|:--------:|:--------:|:--------:|:--------:|:--------:|:--------:|
LD+KD+GI imitation (MSE) | 20.7 | 27.1 | 31.4 | 32.9 | 35.2 | 36.3 | 36.7 | 37.7 | 41.8 | 42.1 | 42.3 | 42.4 |
LD+KD+GI imitation (CI loss) | 25.5 | 30.4 | 33.8 | 35.3 | 36.7 | 37.1 | 37.8 | 37.8 | 42.2 | 42.3 | 42.4 | 42.6 |

Therefore, if you want to show the convergence speed, you'd better draw some AP curves to exhibit.

4. From Sec. 3.4 "Connection of PCC and KL divergence", if the temperature $T$ is high, minimizing KL divergence between post-normalized features is equivalent to minimizing MSE between post-normalized ones,  and equivalent to PKD. However, the experiment results in the Appendix only show the performance of $T=50$. What would be the performance if lower temperatures were used?

5. In object detection, some feature imitation methods transfer the knowledge within some pre-defined imitation regions, and be validated to be better than distilling on the full feature map. While your method operates on full map. How does PKD compare to these approaches, e.g., Fine-Grained feature imitation [33], GI imitation [9]? More specically, conducting PKD within some pre-defined imitation regions, **not just comparing to the results from the original papers**. I know it will work, but just want to see these relevant experiments in the paper, cuz it's about the universality of this method.

6. Part of the experiments are redundant. You spend lots of space to show the effectiveness of PKD on various detectors, such as RetinaNet with different backbones, FCOS, GFL and TOOD. These one-stage detectors share almost the same framework. It would be better if we could choose the results and leave the space for the more important experiments, see above. BTW, Sec. 4.3.2 " Sensitivity study of loss weight $\alpha$" is not important and can be moved to the Appendix.

7. The inheriting strategy of [16] is unnecessary at all. It is irrelevant to your contribution. What readers really want to see is to what extent your PKD helps improve the distillation performance, not anything else. This problem is mentioned in Weakness 2 either. As I know, some previous works (like FRS) use both feature imitation and logit mimicking (classification KD). One solution is to specify which method your PKD is based on. Since PKD is made for replacing MSE loss, it is not difficult to implement that. For example:

|Method | schedule | AP | AP50 | AP75 | APs | APm | APl |
|:--------:|:--------:|:--------:|:--------:|:--------:|:--------:|:--------:|:--------:|
GFL-Res101 (T) | 1x | 44.9 |
GFL-Res50 (S) | 1x | 40.2 |
FitNet | 1x | xxx
GI imitation | 1x | xxx |
FGFI | 1x | xxx |
FRS | 1x | 43.6 |
FGD | 1x | 41.3 |
||
FitNet + PKD | 1x | xxx
GI imitation + PKD | 1x | xxx |
FGFI + PKD | 1x | xxx |
FRS + PKD | 1x | xxx |
FGD + PKD | 1x | xxx |

And the same goes for the heterogeneous detector pair (choose two typical pairs are enough. One is a detector pair that MSE fails. The other is MSE works). The above two tables are Sec. 4.1 "Main Results", the rest detector pairs can be reported in Sec. 4.2, namely "More Teacher-Student Pairs", and picked the best results of your PKD. (among FitNet+PKD, GI imitation+PKD, FGFI+PKD, FRS+PKD, FGD+PKD ect.)

8. Some important relavant works are missing.

[r1] Chen Y, Chen P, Liu S, et al. Deep structured instance graph for distilling object detectors[C] Proceedings of the IEEE/CVF International Conference on Computer Vision. 2021: 4359-4368.

[r2] Zheng Z, Ye R, Hou Q, et al. Localization distillation for object detection[J]. arXiv preprint arXiv:2204.05957, 2022.

---

> ### Author Response · Authors · 2022-08-02
> **Response to reviewer JKs3. Thanks for the feedback!**
>
> We are encouraged that you think our work is of novelty and effectiveness and appreciate your advice. We would like to address your concerns one by one.
> ### Q1 The fine-tuning experiments
> A1: Thanks for the suggestion. That's not a very accurate statement in lines 158-159. It should be replaced with "Backbone and neck replacement simulates the extreme case of knowledge distillation. Distillation between heterogeneous detectors is possible, but feature magnitude difference needs to be eliminated first."
>
> As detection heads and label assignments are usually different between heterogeneous detectors, people may wonder if FPN feature imitation is still applicable for heterogeneous detectors. And the fine-tuning experiments show that it still makes sense. As shown in Tab 1, the pre-trained backbone and neck of GFL can be used to fine-tune the FCOS head and achieve higher performance. While the main idea of feature-based distillation methods is to match the feature activations of the teacher and the student. So **if a well-trained student (mse(feat_t, feat_s) == 0) can output the same features as those of its teacher, it does achieve higher performance.** This is why FPN feature imitation does make sense although the student-teacher detector pairs are heterogeneous. However, the fine-tuning experiments also show that there is a significant mAP drop when fine-tuning the Retina head with the pre-trained backbone and neck of FCOS. In other words, **if a well-trained Retina student can output the same features as those of its teacher FCOS, there will be a significant performance drop.** Hence, feature magnitude difference needs to be eliminated first.
>
> Based on these observations, we propose to replace MSE with 1-PCC in Sec. 3.3. And results in Tab. 4 confirm PCC makes the feature imitation more efficient. For the homogeneous detectors, the feature magnitude difference is relatively small. But dominant FPN stages and channels can still negatively interfere with the training phase. And solving them by PCC can produce a non-negligible improvement (pls refer to A3 to reviewer#1).
> ### Q2 The implementation details and FGD results
> A2:
>
> **The implementation details**
>
> All the PKD results are based on Fitnet instead of some other methods (no pre-defined imitation regions, no logit mimicking and no hidden tricks). **Full map feature imitation with PCC can already outperform these feature imitation methods.**
>
> **FGD results**
>
> We conduct the experiment on GFL according to the hyper-parameters provided in the original paper of FGD ($\alpha$=1e-3, $\beta$=5e-4, $\gamma$=1e-3, $\lambda$=5e-6, T=0.5 for all the anchor-based one-stage models) and it does get 41.3 AP. Maybe better hyper-parameters could lead to higher AP than 41.3. **Here comes another strength provided by PCC. We can train different networks with relatively uniform hyper-parameters.** It is easy for practical use.
> ### Q3 Convergence speed
> A3: We believe these comparisons are fair because the student detectors are the same. We use a stronger heterogeneous teacher just to emphasize the generality of our PKD. In the end, we want to improve the student's performance as much as possible. While with a stronger heterogeneous teacher FCOS-ResX101, Retina-Res50 with our PKD under 1x schedule outperforms previous methods under 2x schedule by a large margin. And that's why we distill heterogeneous detectors. Using the same FCOS-ResX101 as a teacher, other methods can only bring a slight performance gain or even degradation.
>
> Even using the same teacher detector, our PKD converges faster than others. As the performance of the Retina-Res101 (38.9) is lower than the distilled student Retina-Res50 (39.6), which will affect the further improvement of the student's performance. We choose GFL-Res101 2x -> FCOS-Res50 2x (**same teacher**). Following your advice, we'll draw the AP curves to exhibit:
> |Method / epoch|2|4|6|8|10|12|14|16|18|20|22|24|
> |-|-|-|-|-|-|-|-|-|-|-|-|-|
> |FGD|27.2|32.3|35.3|36.0|37.3|38.0|37.9|38.7|42.2|42.5|42.6|42.8|
> |Ours|34.0|36.9|38.4|39.4|40.0|40.3|40.6|40.6|43.1|43.3|43.3|43.5|
> ### Q4 KL divergence with different T
> A4: Please refer to A3 in the comments to all the reviewers.
> ### Q5&7 Comparison with other methods
> A5: **Only full map FPN feature imitation is used in PKD**. Please refer to A1 and A2 in the comments to all the reviewers for details. Forgive me for not having enough time to search for the best loss weight and only comparing PKD with FitNet, GT Mask, FRS and FGD. All the hyper-parameters are copied from the original paper and the only hyper-parameter - distillation loss weight in PKD equals to 10 for all experiments. We will complete it in the coming month.
> ### Q6 About redundant experiments
> A6: We'd like to move some experiments to the Appendix. However "Sensitivity study of loss weight" is important. PKD is not sensitive to the only hyper-parameter. So it is easy for practical use.
> ### Q8
> A7: We will cite these relevant works in the paper.

---

> > ### Comment · Reviewer_JKs3 · 2022-08-04
> > **Response to Authors**
> >
> > Weakness 1,2,4,5,7 are addressed by the authors.
> >
> > The authors response to Weakness 6 is somewhat acceptable, and is not a big issue.
> >
> > The explanation of Weakness 3 is not convincing to me. Sec 4.3.3 is about the convergence speed, and is not about showing the generality of PKD. Therefore, same teacher and same student are required. And the results of table 5 are from the original papers. If Retina-Res101 $\rightarrow$ Retina-Res50 PKD 1x can only achieve the comparable performance to previous methods. I would recommend you to remove the table 5, and replace with the AP curve as your table shown above. Of course, you can also add one more teacher-student pair to make the figure of the AP curve better.

---

> > > ### Author Response · Authors · 2022-08-05
> > > **Response to reviewer JKs3**
> > >
> > > Thanks for the suggestion. We'll use the same teacher-student detector pair to show the convergence speed as Sec 4.3.3 is not about the generality of PKD. And table 5 will be replaced with the AP curve as the table shown above. Following your advice, we'll add the result of FRS to make the figure better.

---

> ### Comment · Reviewer_JKs3 · 2022-08-05
> **My rating is changed**
>
> My rating score in changed from 4 to 6 now.
> The soundness is changed from 1 (poor) to 3 (good).
>
> The rest parts of concern are minor.
>
> 1. the property analysis of $L_{FPN}$ is necessary.
>
> 2. the main results have to be completed. See weakness 7.

---

> > ### Author Response · Authors · 2022-08-05
> > **Response to reviewer JKs3. Thanks for the reply.**
> >
> > Thanks for the reply. We will keep on answering questions and revising our manuscript.
> > ### The property analysis of $L_{FPN}$
> > The Pearson correlation coefficient (r) is essentially a normalized measurement of the covariance, such that the result always has a value between −1 and 1. Hence $L_{FPN} = 1 - r$ always has a value between 0 and 2. It focuses on the linear correlation between features of the teacher and the student and relaxes constraints on the magnitude of the features. Let $\mathbf{X^c}, \mathbf{Y^c} \in R^{nhw}$ denote the c-th channel of the student's and teacher's features, and omit c for clarifying. Then we get $nhw$ data points ($X_i, Y_i$). $L_{FPN} = 0$ implies that all data points lie on a line for which $Y$ increases as $X$ increases. Hence the student is well-trained. And vice versa for $L_{FPN} = 2$. A value of 1 implies that there is no linear dependency between the features of the student and the teacher.
> >
> > Following your advice, we'd like to add these to the paper.
> >
> > ### The main results
> > We will complete it as soon as possible.

---

### Official Review · Reviewer_96XY · 2022-07-06

**Rating:** 6
**Confidence:** 4
**Soundness:** 3 good
**Presentation:** 4 excellent
**Contribution:** 3 good

**Summary:**

The paper proposes a new KD method for object detection model. They distill the features of FPN by first normalize the output of the feature and then minimize the MSE loss which is equivalant to maximizing the pearson correlation coefficient. The experimental results demonstrate the effectiveness of the proposed method.

**Questions:**

- Is teacher the first line in Fig.1? Besides, the font size is too small in Fig.1.

- Lack of ablation studies of using KL divergence with different T.

- Other questions see cons above.

**Limitations:**

See above.

**Strengths And Weaknesses:**

Pros
- The paper is well written and easy to understand.

- The experimental results well demonstrate the effectiveness of the proposed method.

Cons:
- It seems that conducting MSE loss between features after normalizing the feature is not novel. Many KD papers for image classification add conv-bn layers after the output of the feature map before applying MSE loss, which seems quite similar to the proposed method.

- I am confused about Sec.3.4, which shows that the proposed method equals to using KL divergence with a quite large T. However, when T is large, the KL divergence flatten the two feature maps, and makes the original distribution less important. If this is so, why is the difference of the FPN feature maps between student and teacher still matter?

---

> ### Author Response · Authors · 2022-08-02
> **Response to reviewer 96XY. Thanks for the feedback!**
>
> Thanks for your careful and valuable comments. We will explain your concerns point by point. Any further discussion would be appreciated.
> ### Q1 About the novelty
> A1: In this paper, we investigate why directly minimizing the MSE between features of the teacher and the student leads to sub-optimal results. The **magnitude difference, dominant FPN stages and channels** are identified to account for the failure, which is ignored by previous works. Based on these observations, a natural thought is replacing MSE with 1-PCC (Pearson Correlation Coefficient) to focus on the linear correlation and relax constraints on feature magnitude. **Although the adopted techniques are mostly not novel by themselves, we have new discoveries which are important to the general distillation framework for object detectors. These are both novel and valuable.** Details in Tab. 4 further confirm the three issues do exist and that solving them can lead to much better performance (pls refer to A3 to reviewer#1).
>
> Besides, an adaptive layer (a conv layer or conv-bn layers, $\phi_{adapt}$ in Equ1) is used by many previous works when the dimensionality of the student’s feature is different from that of the teacher’s. Some of them find that even when the dimensionality is the same, the adaptive layer may still be necessary. As the teacher and the student may have significantly different network architectures, their feature spaces may be misaligned. The teacher and the student feature spaces, even when they encode the same semantic information, can still be subject to differences caused by transformations such as rotation and scaling. And directly imitating the teacher’s features will lead to lower performance. For example, there is about a 0.6-0.7 mAP drop without a 3x3 adaptive conv layer in FRS[39]. And **an adaptive layer is adopted to solve this**. However, no adaptive layer is needed in our PKD unless the number of channels mismatches.
>
> In addition, the BN module used in our PKD has **no learnable affine parameters**. Firstly, it will lead to collapse solutions if the adaptive layer is applied to the teacher. And if BN with learnable affine parameters is applied to the student's output, the learnable weight and bias will quickly approximate the variance and mean of the features of the teacher, to find a low-loss solution easily. While variance and mean of different teachers' output tend to be different. So we can't use a relatively uniform loss weight to keep the relative contributions of original and distillation loss remain roughly unchanged. But the unlearnable BN can be applied to both the teacher and the student, and that's exactly what PCC does.
>
> Last but not least, as detection heads and label assignments are usually different between heterogeneous detectors, people may wonder if FPN feature imitation is still applicable for heterogeneous detectors. We simulate the extreme case of knowledge distillation via backbone and neck replacement in Sec. 3.2. And demonstrate that distillation between heterogeneous detectors is possible, but feature magnitude difference needs to be eliminated first. It's both novel and valuable.
>
> **With the help of PCC, our method indeed mimics features more accurately and achieves state-of-the-art performance. It's simple yet effective. We hope our method could serve as a solid baseline and help ease future research in knowledge distillation for object detectors.**
>
> ### Q2 About the temperature of KL divergence
> A2: When T is large, the KL divergence flattens the two feature maps and makes the original distribution less important. However, as shown in Equ 7, we usually multiply the KL divergence by $T^2$ to ensure the relative contributions of the original and distillation loss remain roughly unchanged if the temperature T is changed. It will enlarge the feature difference. That's why the difference in the FPN feature maps between student and teacher still matters.
>
> There is detailed proof in Sec. 3.4, and it can also be verified by the following codes:
> ```
> >>> def norm(f):
> ...	return (f - f.mean(-1)) / (f.std(-1) + 1e-8)
> >>> def kl(f_t, f_s):
> ... 	temp = 50  # high temperature
> ... 	return temp ** 2 * F.kl_div((f_s / temp).softmax(dim=-1).log(), (f_t / temp).softmax(dim=-1), reduction='batchmean')
> >>> def mse(f_t, f_s):
> ... 	return F.mse_loss(f_t, f_s) / 2
> >>> preds_S = torch.arange(10).reshape(1, -1).float()
> >>> preds_T = torch.ones_like(preds_S)
> >>> kl(preds_T, preds_S)
> tensor(4.1234)
> >>> mse(preds_T, preds_S)
> tensor(10.2500)
> >>> kl(norm(preds_T), norm(preds_S))
> tensor(0.4500)
> >>> mse(norm(preds_T), norm(preds_S))
> tensor(0.4500)
> ```
> ### Q3 KL divergence with different T
> A3: Please refer to A3 in the comments to all the reviewers.
> ### Q4 About Fig. 1
> A4: The first line in Fig.1 is the teacher. I'm sorry for the small font size. It's really hard to show all three observations in one figure. More visualization is provided in the Appendix. We will carefully revise the manuscript to make it easier to read.

---

> > ### Comment · Reviewer_96XY · 2022-08-10
> > **After Response**
> >
> > Thanks for your response. My concerns are well addressed in this response. Thus, I increase my score to accept this paper.

---

### Official Review · Reviewer_aChz · 2022-07-11

**Rating:** 6
**Confidence:** 4
**Soundness:** 3 good
**Presentation:** 3 good
**Contribution:** 3 good

**Summary:**

In this submission, the authors begin with a clear motivation that the feature value magnitude of the teacher and student is different in pixels, channels and stages for different detector pipelines with different backbones. Motivated by this, the authors propose the Pearson correlation coefficient based approach to tackle the magnitude difference of the distilled features between both homogeneous and heterogeneous detectors. Experiments and ablation studies on the COCO dataset evidence the effectiveness of the proposed method with part of the experiment settings, however, there are some results that should be clarified. Please refer to the weaknesses part.

**Questions:**

I am open to changing my mind if the authors addressed my concerns above.

**Limitations:**

No. They claim there is no potential negative social impact of this work to their knowledge.

**Strengths And Weaknesses:**

**Strengths**
+ The submission is well-written and easy to follow.
+ The motivation is clearly explained and evidenced by comprehensive experiments with promising results under some teacher-student pairs settings.

**Weaknesses**

However, there are still some concerns to be addressed:

1. The normalization process proposed in lines 166-175 is somehow vague. Please clarify it in more details, like $\hat{s} = ?$.

2. To my understanding, the proposed PCC is able to normalize the magnitude of the feature map so that it can compute the loss on the pre-normalized feature maps. Here comes a question, how will other feature-based distillation methods work with the normalized feature maps?

3. As for the 'MSE' in Table 4, could you please clarify whether the MSE works on the pre-normalized feature or the post-normalized feature?

4. Table 2 shows the results of different distillations in the homogeneous setting. The SOTA result is the proposed PKD with inheriting strategy. But, the inheriting strategy which initializes the student with the teacher’s neck and head parameters and gets better results can be applied to all other methods as well in Table 2. To this end, it is unfair to claim PKD is the SOTA.

5. Table 3 also misses some other feature-based distillation methods between heterogeneous detectors, like FKD[37] to further evidence the effectiveness of the PKD.

---

> ### Author Response · Authors · 2022-08-02
> **Response to reviewer aChz. Thanks for the feedback!**
>
> Thanks for your careful and valuable comments. We will explain your concerns point by point. Any further discussion would be appreciated.
> ### Q1 The unclear sentence
> A1: Thanks for the suggestion, we will declare it in more detail in the paper. Line 166-175 provides the definition of $ \hat{s_i} $. In other words, for a mini-batch of FPN outputs (nchw), let $\mathbf{s^{(c)}} \in \mathbb{R}^{nhw}$ be the c-th channel of FPN output features and omit c for clarity. Then the normalized values $ \hat{s_i} $ can be formulated as :
> $$ \begin{aligned}
> \hat{s_i} = \frac{s_i - \mu} {\sigma + \epsilon},  \quad \mu = \frac{1}{nhw} * \sum_{j = 1}^{nhw} s_j , \quad \sigma =  \sqrt{\frac{\sum_{j = 1}^{nhw} (s_j - \mu)^2}{nhw - 1}}
> \end{aligned}$$
> As different elements of the c-th channel of FPN outputs, at different locations, are produced by the same convolution kernel, it is intuitive to normalize the features in this way.
>
> ### Q2 How will other feature-based distillation methods work with PKD
> A2: We believe full map feature imitation with PKD can already outperform those methods with pre-defined imitation regions and hand-crafted designs. And combining PKD with these feature imitation methods can further improve their performance. Please refer to the A1 in the comments to all the reviewers for more details.
>
> ### Q3 Whether the MSE works on the pre-normalized feature or the post-normalized feature
> A3: In Sec. 3.3 we propose a new loss function for feature imitation in object detection. We first conduct feature normalization and then calculate the MSE loss between two normalized features. Mathematically, it is equivalent to firstly calculate the Pearson Correlation Coefficient $r$ between two original feature vectors, and then use 1-$r$ as the new feature imitation loss.
>
> In Sec. 1, we first find that the magnitude difference, dominant FPN stages and channels can negatively interfere with the training phase. Moving forward, results in Tab. 4 further confirm the three issues do exist and that solving them can lead to much better performance (ref to lines 255-263). For the 1st pair (FCOS-ResX101 -> Retina-Res50), the feature magnitude of the teacher and the student is quite different (pls refer to Fig. A in the Appendix). Feature imitation with MSE enforces overly strict constraints and does harm to the student. While for the 2nd and 3rd pairs, the feature magnitude of teacher and student is relatively closed, but the dominant FPN stages and channels can still lead to sub-optimal. And solving these two issues (by using PCC) can bring non-negligible performance gains.
>
> For instance, for the 2nd pair (Gfl-Res101 -> FCOS-Res50), FPN stages 'P5' and 'P6' of both the teacher and the student are less activated than FPN stages 'P3' and 'P4' (pls refer to Fig. B in the Appendix). And it is a common case among different detector pairs. While FPN stages with larger values could dominate the gradient of the distillation loss, which will overwhelm the effects of other features in KD. And large objects are usually assigned to high-level features and vice versa. Hence, as shown in Tab. 4, when PCC is used as the distillation loss (the 2nd issue is solved), the mAP of the student detector for large objects is much higher than that when MSE is used. More visualizations are listed in the Appendix.
> ### Q4 About the inheriting strategy
> A4: We believe these comparisons are fair. **PKD w/o inheriting strategy is still SOTA (even better than SOTA methods w/ inheriting strategy in some teacher-student pairs)**. Besides, PKD can produce a more obvious performance improvement when the student and the teacher detectors are heterogeneous. Please refer to A2 in the comments to all the reviewers for details.
> ### Q5 About other feature-based distillation methods such as FKD[37]
> A5: Here are some more experimental results about FKD[37], FGD[34] and ours on MS COCO2017. Results of FKD and FGD are borrowed from FGD. Note that Retina-ResX101 and Retina-Res50 are regarded as heterogeneous detector pairs in FGD. ($\dagger$ means using inheriting strategy.)
> |detector|mAP|
> |-|-|
> |Retina-ResX101 (T)|40.8|
> |Retina-Res50 (S)|37.4|
> |FKD|39.6|
> |FGD |40.4 |
> |FGD $\dagger$|40.7|
> |**Ours**|**40.8**|
> |RepPoints-ResX101-DCN (T)|44.2|
> |RepPoints-Res50 (S)|38.6|
> |FKD|40.6|
> |FGD|41.3|
> |FGD $\dagger$|42.0|
> |**Ours**|**42.3**|
>
> We would like to add these results to the paper.

---

> > ### Comment · Reviewer_aChz · 2022-08-09
> > **Change the score to 6**
> >
> > Thank you for the reply and additional experiments. The paper is well-motivated, thus I change the score.

---

### Author Response · Authors · 2022-08-02
**Response to all the reviewers. Thanks for the feedback!**

Thanks for your careful and valuable comments. We will explain the concerns in common point by point. Any further discussion would be appreciated.
### Q1 How will other feature-based distillation methods work with PKD
A1: We believe full map feature imitation with PKD can already outperform those methods with pre-defined imitation regions and hand-crafted designs. Besides, PCC is a measure of linear correlation between two features. It is essentially a normalized measurement of the covariance, such that the result always has a value between −1 and 1. However, the boundness will be unpredictable when conducting PKD with a specific imitation mask (e.g. based on FRS[39] and FGD[34]).

**Recaps**:

1. Feature imitation and logit mimicking with FRS uses the aggregated classification score, takes maximum operation in channel direction, of the classification head output as a weighted score mask, to guide the distillation of the FPN and classification head.

2. Focal distillation in FGD forces the student to focus on the teacher’s pixels and channels with large absolute mean values. Global distillation in FGD rebuilds the relation between different pixels and transfers it from teachers to students through GcBloack [40].

Forgive me for not having enough time to search for the best loss weight. All the hyper-parameters are copied from the original paper and the only hyper-parameter - distillation loss weight in PKD equals to 10 for all experiments. We will complete it in the coming month. Here are the results:
GFL Res101 2x -> GFL Res50 1x
|Method|mAP|Remark|
|-|-|-|
|Baseline|40.2||
|FitNet|41.3|Feature imitation with MSE loss|
|GT Mask|41.4|
|FRS feature imitation|43.2||
|FRS feature imitation + logit mimicking|43.6||
|FGD focal|41.8||
|FGD global|40.2||
|FitNet + PKD|43.3|Replace MSE loss with PKD|
|GT Mask + PKD|43.0|Calculate PCC within the GT areas|
|FRS feature imitation + PKD|43.4|Add PKD loss|
|FRS feature imitation + PKD|43.4|Replace MSE loss with PKD|
|FRS feature imitation + logit mimicking + PKD|43.6|Add PKD loss|
|FGD global + PKD|43.3|Add PKD loss|
|FGD global + PKD|43.3|Replace MSE loss with PKD|

Results show that combining PKD with these feature imitation methods can further improve their performance. Besides, we don't combine PKD with FGD focal distillation as focal distillation forces the student to focus on the teacher’s pixels and channels with large absolute mean values. The feature with a larger value is given a larger weight, which seems not to agree with the argument that "The small gradients produced by less activated features can be drowned into the large gradients produced by highly activated ones, thus limiting further refinement." in [11], [38] and us.

Furthermore, **our PKD can produce a more obvious performance improvement when the student and the teacher detectors are heterogeneous**. Please refer to A2 and A3 for more details.

### Q2 About the inheriting strategy
A2: Following FGD[34], we put both results w/ and w/o inheriting strategy in Tab. 2. Inheriting strategy does bring another improvement. But **PKD w/o inheriting strategy is still SOTA (even better than SOTA methods w/ inheriting strategy in some teacher-student pairs)**. Detailed results are shown in Tab 2. We take two SOTA methods FRS[39] and FGD[34] as an example. ($\dagger$ means using inheriting strategy.)
|detector|mAP|
|-|-|
|Retina-ResX101 (T)|40.8|
|Retina-Res50 (S)|37.4|
|FRS|40.1|
|FGD|40.4|
|FGD $\dagger$|40.7|
|**Ours**|**40.8**|
|FCOS-Res101 (T)|41.2|
|FCOS-Res50 (S)| 39.1|
|FRS|42.2|
|FGD|42.3|
|FGD $\dagger$|42.8|
|**Ours**|**42.8**|
|RepPoints-ResX101-DCN (T)|44.2|
|RepPoints-Res50 (S)|38.6|
|FGD|41.3|
|FGD $\dagger$|42.0|
|**Ours**|**42.3**|
### Q3 KL divergence with different T
A3: We conduct experiments on Retina-ResX101 -> Retina-Res50 using KL divergence with different T. The results are as follows:
|T|mAP|
|-|-|
|0.1|nan|
|1|40.7 (+3.3)|
|10|40.8 (+3.4)|
|50|40.9 (+3.5)|

The results demonstrate that feature normalization is the key to conducting feature imitation.

[11] Distilling object detectors via decoupled features. CVPR2021

[38] Decoupled knowledge distillation. CVPR2022

[40] Non-local networks meet squeeze-excitation networks and beyond. ICCVW2019

---

> ### Comment · Reviewer_JKs3 · 2022-08-03
> **Question about the above tab Q1**
>
> What are the imitation loss weights you used for FitNet and GT mask? Because I searched multiple loss weights for them, and never surpassed 41.0.

---

> > ### Author Response · Authors · 2022-08-03
> > **Response**
> >
> > **For FitNet**
> >
> > FPN stages 'P3'-'P6' are used in all the experiments with pkd and vanilla mse. No adaptive layers are needed.
> > MSE loss weight equals 50.
> > Here is the Pseudo code:
> > ```
> > def loss(preds_S, preds_T):
> >     loss = 0.
> >     for ps, pt in zip(preds_S, preds_T):
> >         loss += F.mse_loss(ps, pt ) / 2
> >     return loss * self.loss_weight
> > ```
> >
> > **For GT Mask**
> >
> > FPN stages 'P3'-'P7' are used in other experiments as it is adopted by FRS, FGD and many other methods. But the first 4 FPN stages are enough for PKD. (Maybe the information in stage 'P7' is not so important as that in the first 4 stages, as it's usually the down-sampled results.)
> > Loss weight equals 10 and I didn't search for the loss weight. Maybe a larger loss weight can lead to better performance.
> > Here is the Pseudo code:
> > ```
> > def loss(feat_T, feat_S):
> >     feat_T, feat_S = feat_T.unsqueeze(0), feat_S.unsqueeze(0)
> >     return F.mse_loss(feat_T, feat_S, reduction='sum'), feat_T.numel()
> >
> > def gt_mask_loss(preds_S, current_data, preds_T):
> >     gt_bboxes, metas = current_data
> >     loss = 0.
> >     for layer_ind, (pred_S, pred_T) in enumerate(zip(preds_S, preds_T)):
> >         assert pred_S.shape[-2:] == pred_T.shape[-2:]
> >         N, C, H, W = pred_S.shape
> >
> >         loss_per_stage = 0.
> >         for i in range(N):
> >             new_boxx = torch.ones_like(gt_bboxes[i])
> >             new_boxx[:, 0] = gt_bboxes[i][:, 0] / metas[i]['img_shape'][1] * W
> >             new_boxx[:, 2] = gt_bboxes[i][:, 2] / metas[i]['img_shape'][1] * W
> >             new_boxx[:, 1] = gt_bboxes[i][:, 1] / metas[i]['img_shape'][0] * H
> >             new_boxx[:, 3] = gt_bboxes[i][:, 3] / metas[i]['img_shape'][0] * H
> >
> >             wmin = torch.floor(new_boxx[:, 0]).int()
> >             wmax = torch.ceil(new_boxx[:, 2]).int()
> >             hmin = torch.floor(new_boxx[:, 1]).int()
> >             hmax = torch.ceil(new_boxx[:, 3]).int()
> >
> >             tmp_loss, cnt = 0., 0
> >             for j in range(len(gt_bboxes[i])):
> >                 l, c = loss(pred_T[i, :, hmin[j]:hmax[j]+1, wmin[j]:wmax[j]+1],
> >                                  pred_S[i, :, hmin[j]:hmax[j]+1, wmin[j]:wmax[j]+1])
> >                 tmp_loss += l
> >                 cnt += c
> >             loss_per_stage += tmp_loss / cnt
> >         loss += loss_per_stage / N
> >
> >     return loss * self.loss_weight
> > ```
> >
> > Here is the training log:
> > |Method / epoch|1|2|3|4|5|6|7|8|9|10|11|12|
> > |-|-|-|-|-|-|-|-|-|-|-|-|-|
> > |MSE|7.7|20.2|27.3|30.2|32.5|34.0|35.6|36.1|40.6|40.7|41.0|41.3|
> > |GT Mask|16.5|24.3|28.6|29.0|33.8|34.6|35.1|35.9|40.5|41.0|41.0|41.4|
> >
> > The configs and codes are based on [mmdet](https://github.com/open-mmlab/mmdetection/blob/master/configs/gfl/gfl_r50_fpn_1x_coco.py) and [mmrazor](https://github.com/open-mmlab/mmrazor)

---

### Author Response · Authors · 2022-08-08
**Looking forward to further discussions!**

Dear Reviewers aChz and 96XY,

We were wondering if our response and revision have addressed all your concerns. As we noted in our previous responses, we conducted additional experiments and attempted to address all of your concerns. In the remaining two days of the rebuttal period, we would appreciate it if you could re-evaluate our submission, or kindly let us know whether you have any other questions, so that we can still have time to respond and address them. We are looking forward to discussions that can further improve our current manuscript. Thanks!

Best regards,

The Authors

---

### Meta-Review · Area_Chair_krtu · 2022-08-26

**Recommendation:** Accept
**Confidence:** Certain

**Metareview:**

This paper proposes a novel knowledge distillation method for object detection. As the feature value magnitude of the teacher and student is different, a new loss function for feature imitation is introduced by conducting feature standardization and calculating the MSE loss, which  is equivalent to calculate the Pearson Correlation Coefficient between two features. Experimental results demonstrate the effectiveness of the proposed method. After an in-depth discussion between the authors and reviewers, the concerns have been well addressed. All the reviewers recommend acceptance. Considering that the overall quality is clearly above the bar, the paper should be accepted for publication. The AC strongly urges the authors to consider all the comments in preparing the final version.

**Award:**

No

---

### Decision · Program_Chairs · 2022-09-14

Accept